# SARS-CoV-2 RNAemia and Disease Severity in COVID-19 Patients

**DOI:** 10.3390/v15071560

**Published:** 2023-07-16

**Authors:** Merlin Jayalal Lawrence Panchali, Choon-Mee Kim, Jun-Won Seo, Da-Young Kim, Na-Ra Yun, Dong-Min Kim

**Affiliations:** 1Department of Internal Medicine, Chosun University College of Medicine, Gwangju 61452, Republic of Korea; lpmjlal@gmail.com; 2Premedical Science, College of Medicine, Chosun University, Gwangju 61452, Republic of Korea; choonmee@chosun.ac.kr (C.-M.K.); kaist-105@hanmail.net (J.-W.S.); dayz02@hanmail.net (D.-Y.K.); shine-0222@hanmail.net (N.-R.Y.)

**Keywords:** RNAemia, SARS-CoV-2, COVID-19, severity, logistic regression analysis

## Abstract

Objective: The clinical implications of SARS-CoV-2 RNA viremia in blood (RNAemia) remain uncertain despite gaining more prognostic implications for coronavirus disease 2019 (COVID-19). However, the clinical relevance of SARS-CoV-2 RNAemia has not been well documented. Methods: We conducted a cohort study on 95 confirmed COVID-19 patients and explored the prospects with evidence of SARS-CoV-2 RNAemia in association with various clinical characteristics. We performed reverse transcription-polymerase chain reaction and studied the risk factors of SARS-CoV-2 RNAemia using logistic regression analysis. Results: The presence of SARS-CoV-2 RNAemia in critical or fatal cases was the highest (66.7%), followed by severe (12.5%) and mild to moderate (1.7%) in admission samples. SARS-CoV-2 viral RNAemia was detected on admission and 1st week samples; however, RNAemia was not detected on the samples collected on the second week post-symptom onset. Multiple regression analysis showed that the severity of the disease was an independent predictor of RNAemia (*p* < 0.021), and the Kaplan–Meier survival curve estimated an increased mortality rate in SARS-CoV-2 RNAemia cases (*p* < 0.001). Conclusions: Our study demonstrated that SARS-CoV-2 RNAemia is a predictive risk factor for clinical severity in COVID-19 patients. Hence, we showed that blood RNAemia might be a critical marker for disease severity and mortality.

## 1. Introduction

Severe acute respiratory syndrome coronavirus 2 (SARS-CoV-2), which emerged in Wuhan, China, in mid-December 2019, is now widespread in over 200 countries and has infected more than 765.9 million people, causing nearly 7 million deaths as of 17 May 2023 [1]. Coronavirus disease 2019 (COVID-19), caused by SARS-CoV-2 manifesting a mild upper respiratory tract infection, is also able to cause severe lower respiratory tract infections, including pneumonia and acute respiratory distress syndrome in a variable number of patients [2,3]. Despite the clinical importance of viremia on disease progression and the pathogenesis of COVID-19, only a few studies have examined the importance of viral load in peripheral blood [4]. One study reported that symptomatic and asymptomatic patients had similar viral loads in respiratory specimens, indicating that disease severity may not objectively depend on the viral load in respiratory specimens [5]. In another study, SARS-CoV-2 viral RNA was detected in 15% of the peripheral blood samples of COVID-19 patients [6]. SARS-CoV-2 viral RNA kinetics in blood have been reported previously; however, it has been poorly understood weather SARS-CoV-2 RNAemia plays a role in disease severity or mortality in follow-up studies in a prospective manner [7,8,9]. Another recent study also exposed that the levels of SARS-CoV-2 RNAemia correlate with disease severity, but the follow-up samples were not studied [10]. Its been reported that SARS-CoV-2 RNAemia lasted for a median of 7 days in a study but they failed to classify the patients according to disease severity [11]. Another study reports the diagnostic sensitivity between the dPCR and qPCR of SARS-CoV-2 RNAemia patients but failed to report disease severity and mortality [12]. A recent study confirms that the RNAemia of patients was the same for patients with and without antiviral therapy [13].

In this study, we quantitatively assessed the dynamics of blood SARS-CoV-2 RNAemia kinetics in 95 patients with clinically confirmed COVID-19. Moreover, using logistic regression analysis, we also found that the presence of SARS-CoV-2 RNAemia was a predictor of mortality.

## 2. Materials and Methods

### 2.1. Participants

We performed a cohort SARS-CoV-2 RNAemia study on 95 patients with confirmed COVID-19 from February 2020 to May 2021 at the Chosun University Hospital, South Korea. All patients were clinically confirmed to be SARS-CoV-2 positive using diagnostic methods such as real-time polymerase chain reaction, cell culture, or a >4-fold increase or seroconversion in terms of the SARS-CoV-2 antibody titer. The initial diagnosis was carried out with respiratory samples. In addition, 24 serum/plasma samples collected from healthy subjects, with no clinical symptoms, no history of antibody detection of SARS-CoV-2, and no history of contact with confirmed COVID-19 patients were used as negative controls.

This study was approved by the Institutional Review Board (IRB) of Chosun University Hospital (IRB 2020-02-011-003), Korea. All research work was performed in accordance with relevant guidelines and regulations set by Chosun University.

### 2.2. Sampling and RNA Extraction

Serum/plasma samples were collected from the peripheral blood obtained from patients, and 200 μL of each sample was used for RNA extraction. In addition, self-collected sputum samples from the patients were diluted in phosphate-buffered saline (PBS), mixed, and centrifuged (200× *g*, 1 min), and the supernatant was subjected to RNA extraction. Nasopharyngeal swabs were collected directly by a physician using commercial UTM™ kits containing 1 mL of viral transport medium (Noble Bio, Seoul, Republic of Korea) and samples were used for RNA extraction. The viral RNA was extracted using a fully automated instrument (PCL, Seoul, Republic of Korea) with the Real-prep Viral DNA/RNA Kit (Biosewoom, Republic of Korea).

### 2.3. Real-Time Reverse Transcription-Polymerase Chain Reaction (qRT-PCR) for SARS-CoV-2 Detection

For the qRT-PCR assay of the *NP* (encoding nucleocapsid protein) gene, primers and probes were designed in-house, nCov-NP_572F (5′-GCAACAGTTCAAGAAATTC-3′), nCov-NP_687R (5′-CTGGTTCAATCTGTCAAG-3′), and nCov-NP_661P (5′-FAM-AAGCAAGAGCAGCATCACCG-BHQ1-3′). Thermal cycling was performed as follows: 50 °C for 10 min for reverse transcription, one cycle of 95 °C for 30 s for pre-incubation, 95 °C for 5 s at 57 °C for amplification, and 45 cycles for data detection. For target genes *E* (encoding envelope protein) and *RdRp* (encoding RNA-dependent RNA polymerase), the Kogene Kit (Kogene Biotech Co., Ltd., Seoul, Republic of Korea) and SD Kit (SD Biotechnologies Co., Ltd., Seoul, Republic of Korea) were used, and the amplification was performed according to the manufacturer’s specifications. For the *NP* target, qRT-PCR was performed in an Exicycler^TM^ 96 Real-Time Quantitative Thermal Block (Bioneer, Daejeon, Republic of Korea), and for Kogene and SD kits, the CFX96 Touch™ Real-Time PCR Detection System (Hercules, CA, USA) was used. Cycle threshold (C_t_) values were set to ≤40 for the reference gene and were assumed to denote a positive result.

### 2.4. Cell Culture

For the identification of SARS-CoV-2 viral isolation, serum/plasma samples were inoculated into a monolayer of Vero E6 cells and incubated in Dulbecco’s modified Eagle’s medium supplemented with 2% fetal bovine serum and a 1× penicillin–streptomycin antibiotic solution (Gibco, Thermo Fisher Scientific, Waltham, MA, USA) in 5% CO_2_ at 37 °C for 3 to 5 days. After two passages, viral proliferation was confirmed based on qRT-PCR with a confirmatory C_t_ value of <20 or indirect immunofluorescence assay (IFA) using in-house SARS-CoV-2 antigen slides [14,15].

### 2.5. Enzyme-Linked Immunosorbent Assay (ELISA)

An indirect ELISA was performed using recombinant SARS-CoV-2 nucleocapsid protein (BIOAPP. Inc., Pohang, Republic of Korea) for serological titers of IgG, IgM, and total antibodies (IgG, IgM, and IgA). Each well of a 96-well ELISA microplate (Thermo Fisher Scientific, Waltham, MA, USA) was coated with 100 µL of 2 µg/mL recombinant SARS-CoV-2 nucleocapsid protein (BIOAPP. Inc.) in carbonate-bicarbonate buffer, followed by overnight incubation at 4 °C. The microplates were washed with PBS supplemented with 0.05% Tween 20 (PBS-T) and blocked with 5% skim milk in PBS-T for 2 h at 37 °C. After washing, the serum samples were diluted 100-fold with blocking buffer and incubated at 37 °C for 2 h. The plates were rewashed, and a secondary antibody (horseradish peroxidase-conjugated goat anti-human IgG antibody (1:6000, Invitrogen, Thermo Fisher Scientific, Cat A18805), anti-human IgM antibody (1:3000, Invitrogen, Thermo Fisher Scientific, Cat 31415), or anti-human total-antibody antibody (1:40,000; Thermo Fisher Scientific, Cat 31418 Waltham, MA, USA) was added, and the plate was incubated again at 37 °C for 1 h. After washing, 50 µL of the 3,3′5,5′-tetramethylbenzidine substrate (Sigma-Aldrich, St. Louis, MO, USA) was added at room temperature (20–30 °C) and incubated for 30 min in the dark. The reaction was stopped with 25 µL of 1 M H_2_SO_4_, and the optical density at 450 nm (OD_450_) was measured using an Epoch™ 2 microplate spectrophotometer (VT, USA). The cutoff values were determined by calculating the mean OD_450_ plus 3-fold standard deviation of the negative serum samples. Thus, the observed cutoffs for IgG, IgM, and total antibodies were 1.1, 0.5, and 0.8, respectively. When the OD of a patient sample was greater than the calculated cutoff, it was considered positive for SARS-CoV-2.

### 2.6. Indirect Immunofluorescence Antibody Assay

For indirect IFA, SARS-CoV-2 samples obtained from the Korea Disease Control and Prevention Agency were used to infect Vero E6 cells. To develop antigen slides of SARS-CoV-2, 3-day-infected cells were cultured on Teflon-coated welled slides at 37 °C with 5% CO_2_ overnight. The slides were then fixed with 80% acetone. Two-fold serial dilutions starting from 1:16 of the patient serum were added and reacted with SARS-CoV-2 viral antigens for 30 min at 37 °C in a moist chamber. The slides were washed and further incubated with 1:400 diluted secondary antibody (fluorescein isothiocyanate-conjugated anti-human IgM and IgG; MP Biomedicals, OH, USA). Then, the slides were observed using a fluorescence microscope (Olympus IX73, magnification: 400×) after dispensing the mounting solution (VECTOR Laboratories) [16]. Using clinical samples of 15 healthy subjects, the IFA antibody titer cutoff was established as ≥1:32.

### 2.7. Statistical Methods

Statistical analyses were performed using MedCalc 20·013 software (Ostend, Belgium) and IBM SPSS Statistics for Windows, version 26.0. (IBM Corp., Armonk, NY, USA). Quantitative variables are presented as the mean ± standard deviation for normally distributed variables. Means were compared using *t*-tests for continuous variables. Coefficients of determination (*R*^2^) were computed using linear regression analysis, which was used for multiple correlation analysis (MedCalc). *p*-values comparing COVID-19 patients with evidence of RNAemia to patients without RNAemia were calculated using the Mann–Whitney U test or Fisher’s exact test, as appropriate.

To determine the 30-day mortality rate, Kaplan–Meier survival analysis was performed in the SARS-CoV-2 RNAemia and non-RNAemia groups. To investigate the correlation of predictive risk factors for RNAemia along with other variables, univariate and multiple logistic regressions were performed. For multiple subgroups, with RNAemia as a predictor, the chi-square test was performed. Viral load comparisons were analyzed using the non-parametric Kruskal–Wallis test followed by the Mann–Whitney U test. Statistical significance was set at *p*  <  0.05 [17].

## 3. Results

### 3.1. Clinical and Demographic Characteristics of Patients

This study included 95 patients with clinically confirmed COVID-19, admitted and treated at a single tertiary hospital (Chosun University Hospital, Gwangju, Republic of Korea) between February 2020 and May 2021. The median age of the patients was 64 ± 18.7 years, and the percentages of men and women were 48% and 52%, respectively. Moreover, among patients with underlying comorbidities (54%), 40% had hypertension and 23% had diabetes mellitus. The detailed characteristics are presented in Table 1. On admission, several patients had symptoms such as fever (18%), cough (17%), headache (5%), chills (13%), sore throat (7%), and myalgia (11%). Additionally, considering the treatment scenario, 51% of patients underwent supplemental oxygen with 24% with high oxygen flow, whereas 15% required mechanical ventilation. Approximately 44% of the patients received antiviral treatment, and 31% underwent steroid therapy concurrently (Table 1).

### 3.2. Biochemical and Laboratory Characteristics of Patients

To determine the diagnostic characteristics and the presence of viral RNAemia, the patients were categorized as asymptomatic, mild to moderate, severe, and critical or fatal according to the Sixth Revised Trial Version of the Novel Coronavirus Pneumonia Diagnosis and Treatment Guidance [18]. The percentages of white blood cells in severe and critical or fatal cases were elevated along with other biochemical features. In contrast, the lymphocyte count decreased in severe and critical or fatal cases, as shown in Table 2.

### 3.3. Viral RNAemia in Serum/Plasma Samples

Serum/plasma samples were analyzed for the presence of the viral RNA of SARS-CoV-2 using *NP* as the target gene. SARS-CoV-2 RNAemia was not detected in asymptomatic patients during the entire study period. In the mild-to-moderate category, RNAemia was detected in the on admission samples (2%) and in the week 1 hospital admission samples (6%). The proportion of RNAemia in severely ill patients was 13% in both the on admission and week 1 samples; however, no RNAemia was detected in week 2 samples collected after hospitalization. In contrast, a substantial proportion of RNAemia was detected in critically ill or fatal patients with COVID-19 (67% of on admission samples), and the viral RNA decreased to 18% in the week 1 (Figure 1). However, none of the samples from week 2 exhibited RNAemia. None of the 12 patients with RNAemia had positive cell culture results. The association according to disease severity was significant on admission (*x*^2^(3) = 48.376, *p <* 0.001). Hence, the proportion of SARS-CoV-2 RNAemia was directly correlated with disease severity (Table 3, Figure 1). Similarly, the viral loads of plasma samples of critical or fatal cases were substantially higher in both on admission and week 1 samples than in the other groups of patients. Moreover, statistical analysis was performed for the viral load of SARS-CoV-2 RNAemia in different patient categories, and the results were significant for on admission samples, as listed in Table 3.

### 3.4. Cell Culture of SARS-CoV-2 RNAemia-Positive Serum/Plasma

Cell culture viral proliferation results were monitored using the supernatant of Vero E6 cell lines after two passages at an interval of 5 days via qRT-PCR with a confirmatory C_t_ value of <20 or IFA. None of the 12 patients in critical and fatal category on admission with RNAemia had positive culture results.

The plasma samples were classified according to the Sixth Revised Trial Version of the Novel Coronavirus Pneumonia Diagnosis and Treatment Guidance, and the data were expressed as (%) of RNAemia. *N* represents the number of patients in accordance with patient classification and time. *x* axis represents the number of viral RNAemia patients and *y* axis represents the presence of percentage of RNAemia.

### 3.5. Correlation of RNAemia in Association with Viral Copy Number in the Upper and Lower Respiratory Tract Specimens

Examination of the correlation of RNAemia with viral copy number in the upper and lower respiratory tract specimens showed that the levels of SARS-CoV-2 viral loads were significantly different between the upper and lower respiratory tract specimens (Table 4). For respiratory tract samples, both upper respiratory tract (nasopharynx + oropharynx swab samples) and lower respiratory tract (sputum) viral loads were assayed for their correlation with disease severity. For all upper and lower respiratory tract samples, both *E* and *RdRp* targets were subjected to qRT-PCR to determine the viral load according to the disease classification and sample collection date (Table 4).

On admission, the viral copy number (6.14 × 10^5^) of the upper respiratory tract (nasopharynx + oropharynx samples) was considerably higher than that of the lower respiratory tract (sputum) samples (1.19 × 10^5^) (r = 0.47, *p* < 0.001). For asymptomatic patients, the viral RNA copy number ranged from 6 to 7 × 10^5^ on admission and decreased to 4–5 × 10^4^ in week 1 and gradually disappeared in week 2 after hospitalization. Moreover, in mild to moderate patients, an average of 10 × 10^7^, 10 × 10^6^, and 7–8 × 10^5^ viral RNA copy numbers were found on admission, during week 1, and in week 2, respectively. In contrast, high viral RNA loads on respiratory samples of critical or fatal cases were detected in the on admission samples (3.5–6 × 10^8^). Similarly, high viral loads were observed in both the week 1 and 2 samples of critical or fatal cases compared with those of the other groups (Table 4).

Furthermore, the correlation between on admission RNAemia and on admission upper respiratory tract samples was r = 0.22 (*p* = 0.013), whereas that with lower respiratory tract samples was r = 0.26 (*p* = 0.003). The correlation of week 1 RNAemia with week 1 upper respiratory tract samples was r = 0.22 (*p* = 0.012), whereas that with week 1 sputum samples was r = 0.32 (*p* < 0.001) (Appendix A). Similarly, for the critical or fatal cases, the correlation of RNAemia with on admission and week 1 samples of the upper respiratory tract samples was r = 0.53, *p* < 0.001 and r = 0.60, *p* < 0.001, respectively, and that for the lower respiratory tract samples was r = 0.31, *p* = 0.047 and r = 0.43, *p* = 0.005, respectively (Appendix A). In summary, viral RNAemia was correlated with viral load in respiratory samples.

### 3.6. SARS-CoV-2 RNAemia in Association with Clinical Risk Factors

The clinical correlation between RNAemia and severity was evaluated using baseline risk factors such as age, sex, and other physical and clinical parameters. Univariate logistic regression analysis showed that age, upper respiratory viral copy number, white blood cell count, neutrophil/lymphocyte ratio, and C-reactive protein were significant (*p* < 0.05); thus, these parameters were considered to be risk factors and were further subjected to multivariate logistic regression analysis. Results of the multiple regression analysis showed that RNAemia and age were predictors of mortality (Table 5).

Disease severity and SARS-CoV-2 RNAemia were directly proportional to patient’s older age. Similarly, other clinical features such as C-reactive protein, white blood cell count, neutrophil count, and correlation with upper respiratory tract sample viral loads directly influenced the presence of viral RNAemia and disease severity. Hence, our results confirmed the presence of RNA in blood samples during the initial stage of infection, especially in the older population, which might be a marker of disease severity. The risk factors and mortality rates of patients with and without RNAemia are presented using the Kaplan–Meier curves in Figure 2. Thus, our results predict a high mortality rate in patients with RNAemia in correlation with other risk factors.

*p*-values of the comparison of patients with COVID-19 with evidence of RNAemia to patients without RNAemia were calculated using the Mann–Whitney U test or Fisher’s exact test, as appropriate. *p <* 0.05 was considered significant.

## 4. Discussion

SARS-CoV-2 RNA in blood, known as RNAemia, and its effect on disease severity and fatal clinical outcomes are not fully understood. The fatality rate for COVID-19 varies by age, with 0.3 deaths per 1000 cases among the young (aged between 5 and 17 years old) to 304.9 deaths per 1000 cases in patients aged >85 years [19]. In a previous study, a qualitative viral detection assay in plasma showed correlation with disease severity [20]. Another study, that was performed by adapting the published real-time RT-PCR assays targeting *E* gene showed that RNAemia was present in approximately one-third of patient’s samples [21].

A few studies have reported RNAemia and disease severity in COVID-19 patients with other clinical aspects, including risk factors such as age, smoking, and comorbidities; however, they failed to reflect disease progression and severity correlated with viral RNAemia [22,23,24]. We explored the correlation between RNA viral load and various demographic, laboratory, biochemical, physical, and clinical parameters of 95 patients with clinically confirmed COVID-19. Moreover, we evaluated the viral load from the detectable SARS-CoV-2 RNA in the upper respiratory tract swab and lower respiratory tract samples along with SARS-CoV-2 RNA in plasma in association with other clinical parameters of COVID-19 characteristics and severity. In addition, as Krifors et al. reported the presence of viremia might not affect with the antiviral treatments.

Our study results are consistent with previously published data, where SARS-CoV-2 RNAemia was higher in critically ill patients than in mild to severe patients and very rare in outpatients [25]. The percentage of RNAemia in critical or fatal cases was the highest (67%), followed by severe (13%) and mild to moderate (2%) cases in the on admission samples. Similarly, SARS-CoV-2 RNAemia was detected in the week 1 samples (critical or fatal [18%], severe [13%], and mild to moderate [5%]). However, RNAemia was not detected in any of the samples from week 2 after symptom onset. Furthermore, none of the asymptomatic patients exhibited RNAemia throughout the study period. The viral load of blood samples was lower than that of the upper and lower respiratory tract samples, and considerably higher viral copy numbers were detected in critical and fatal case on admission samples than in the on admission samples of severe and mild to moderate cases. The main strength of our study lies in addressing the presence of RNAemia in the follow-up samples, classified according to the severity of the patients’ illness, ranging from mild to severely ill. However, the main limitation of our study is the sample size. In addition, we cannot eliminate the low number of samples and heterogeneity might contributed to an very wide range of variance.

## 5. Conclusions

To summarize, we discussed the quantitative determination of blood SARS-CoV-2 RNAemia with clinical significance in COVID-19 patients. Using logistic regression analysis, we showed that SARS-CoV-2 RNAemia was a risk factor for disease severity. Simultaneously, this study provides evidence that the presence of RNAemia is associated with a higher mortality rate than that in non-RNAemia cases.

## Figures and Tables

**Figure 1 viruses-15-01560-f001:**
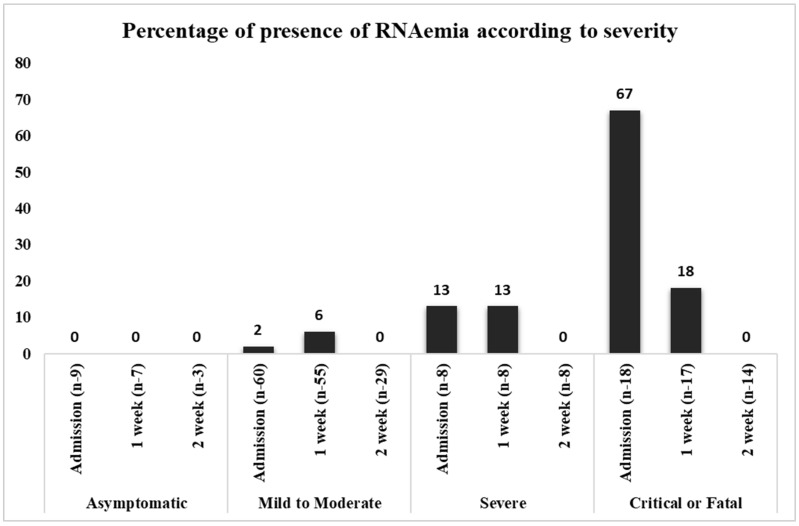
Direct proportion of viral RNAemia in disease severity.

**Figure 2 viruses-15-01560-f002:**
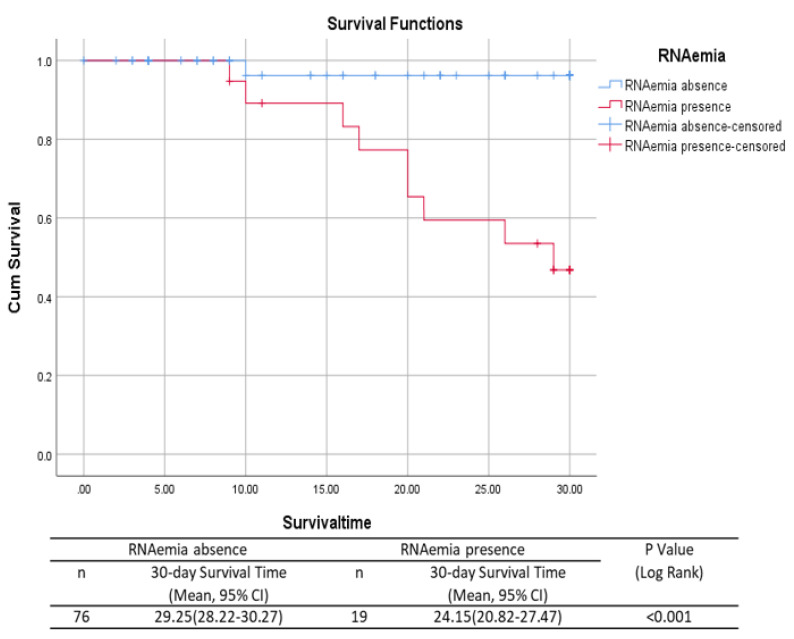
Mortality in patients with and without RNAemia using the Kaplan–Meier curve.

**Table 1 viruses-15-01560-t001:** Clinical characteristics of patients with COVID-19.

Characteristics	Total (*n* = 95)	RNAemia Presence (*n* = 19)	RNAemia Absence (*n* = 76)	*p*-Value
Male, *N* (%)	46 (48%)	9 (47%)	37 (49%)	0.918
Age, mean ± SD	64 ± 18.7	76.8 ± 12	60.8 ± 18.8	<0·001
Comorbidities, *N* (%)	51 (54%)	13 (68%)	38 (50%)	0.150
Cardiovascular disease, *N* (%)	15 (16%)	3 (16%)	12 (16%)	1.000
Diabetes mellitus, *N* (%)	22 (23%)	7 (37%)	15 (20%)	0.134
Hypertension, *N* (%)	38 (40%)	11 (58%)	27 (36%)	0.075
Chronic lung disease, *N* (%)	1 (1%)	1 (5%)	0 (0)	0.200
Cancer, *N* (%)	8 (8%)	1 (5%)	7 (9%)	0.579
Chronic kidney disease, *N* (%)	2 (2%)	1 (5%)	1 (1%)	0.362
Symptoms, *N* (%)				
Fever, *N* (%)	17 (18%)	6 (32%)	11 (15%)	0.082
Cough, *N* (%)	16 (17%)	3 (16%)	13 (17%)	0.891
Headache, *N* (%)	5 (5%)	1 (5%)	4 (5%)	1.000
Chill, *N* (%)	12 (13%)	2 (11%)	10 (13%)	0.757
Sore throat, *N* (%)	7 (7%)	1(5%)	6 (8%)	0.695
Myalgia, *N* (%)	10 (11%)	2 (11%)	8 (11%)	1.000
Treatments				
Supplemental oxygen, *N* (%)	48 (51%)	19 (100%)	28 (37%)	<0.001
High flow oxygen therapy, *N* (%)	23 (24%)	13 (68%)	10 (13%)	<0.001
Mechanical ventilation, *N* (%)	14 (15%)	11 (58%)	3 (4%)	<0.001
Antiviral, *N* (%)	42 (44%)	19 (100%)	23 (30%)	<0.001
Remdesivir, *N* (%)	35 (37%)	17(89%)	18 (24%)	
Kaletra, *N* (%)	6 (6%)	2(11%)	4 (5%)	
Oxyclorin, *N* (%)	1 (1%)	0	1 (1%)	
No antiviral agents, *N* (%)	53 (56%)	0	53 (70%)	
Steroids, *N* (%)	29 (31%)	13 (68%)	16 (21%)	<0.001

Data are expressed as the mean ± SD or *N* (%). The data consisted of 95 patients who participated in this study. *N*: number of patients; SD: standard deviation.

**Table 2 viruses-15-01560-t002:** Biochemical and immunological findings of patients with COVID-19 on admission.

Laboratory Variables	Total (*N* = 95)	Asymptomatic (*N* = 9)	Mild to Moderate (*N* = 60)	Severe (*N* = 8)	Critical or Fatal (*N* = 18)
WBC × 10^9^ per L	6.18 ± 3.08	6.74 ± 2.63	5.44 ± 1.66	7.24 ± 3.53	7.92 ± 5.34
Neutrophils (%)	69.01 ± 14.44	55.99 ± 15.47	65.12 ± 11.85	81.2 ± 7.9	83.06 ± 10.26
Lymphocytes (%)	22.33 ± 12.07	35.12 ± 13.62	24.96 ± 10.34	12.14 ± 5.08	11.71 ± 7.57
Creatinine (mg/dL)	1.00 ± 1.36	0.68 ± 0.10	0.98 ± 1.36	1.78 ± 2.82	0.89 ± 0.45
CRP (mg/dL)	4.47 ± 6.09	0.28 ± 0.33	2.18 ± 3.00	6.19 ± 5.80	11.48 ± 7.88
Procalcitonin (ng/mL)	0.234 ± 0.684	0.030 ± 0.011	0.090 ± 0.108	0.125 ± 0.061	0.873 ± 1.435
Troponin-I (ng/mL)	0.031 ± 0.085	0.001 ± 0.001	0.029 ± 0.093	0.012 ± 0.010	0.056 ± 0.092
AST (U/L)	35.57 ± 26.44	17.49 ± 3.81	31.31 ± 20.17	51.95 ± 50.62	51.52 ± 27.54
ALT (U/L)	25.78 ± 32.21	16.36 ± 7.19	24.91 ± 30.36	51.43 ± 70.78	22.01 ± 8.77

Data are expressed as the mean ± SD or *N* (%). *N*, number of patients; CRP, C-reactive protein; AST, aspartate aminotransferase; ALT, alanine aminotransferase.

**Table 3 viruses-15-01560-t003:** Viral RNA presence in serum/plasma of patients with COVID-19.

	Severity	RNAemia *			Viral Load Mean (SD)		
		*N* (%)	Chi- Square	*p*-Value		Kruskal–Wallis Test *p*-Value	Mann–Whitney U Test *p*-Value
Admission	Asymptomatic ^a^ (*N* = 9)	0 (0)	48.376 ^a^	<0.001	0	<0·001	<0.001 ^b,d^/0.002 ^a,d^/0.009 ^c,d^
	Mild to Moderate ^b^ (*N* = 60)	1 (2)			5.91 × 10^2^ (±4.5 × 10^2^)		
	Severe ^c^ (*N* = 8)	1 (13)			1.29 × 10^2^ (±3.64 × 10^2^)		
	Critical or Fatal ^d^ (*N* = 18)	12 (67)			6.62 × 10^3^ (±1.54 × 10^4^)		
Week 1	Asymptomatic (*N* = 7)	0 (0)	3.115 ^a^	0.374	0	0·360	
	Mild to Moderate (*N* = 51)	3 (6)			1.03 × 10^2^ (±4.23 × 10^2^)		
	Severe (*N* = 8)	1 (13)			1.22 × 10^2^ (±3.45 × 10^2^)		
	Critical or Fatal (*N* = 17)	3 (18)			5.24 × 10^2^ (±1.26 × 10^3^)		
Week 2	Asymptomatic (*N* = 3)	0 (0)	NA		0	NA	
	Mild to Moderate (*N* = 28)	0 (0)			0		
	Severe (*N* = 8)	0 (0)			0		
	Critical or Fatal (*N* = 14)	0 (0)			0		

Data are expressed as the mean ± standard deviation (SD) or *N* (%). *N*: number of patients; NA: not applicable. First sample is on admission, 1st week follow-up sample collected from the 5th to 9th day of admission, 2nd week sample collected from the 12th to 16th day of admission. * RNAemia comparisons among multiple subgroups were performed using the chi-square test. ^a^ represents the cells with the expected count. Viral load comparisons were analyzed using the chi-square test, followed by the non-parametric Kruskal–Wallis test and the Mann–Whitney U test. ^a, b, c,^ and ^d^ represent the patient categories. *p* < 0.05 is considered significant.

**Table 4 viruses-15-01560-t004:** Viral RNA copy numbers in upper and lower respiratory tract specimens.

Participants		Upper Respiratory Tract Viral Load		Lower Respiratory Tract Viral Load	
		*E*-Gene	*RdRp*-Gene	*E*-Gene	*RdRp*-Gene
Asymptomatic	Admission (*N* = 9)	6.14 × 10^5^	7.69 × 10^5^	1.19 × 10^5^	9.32 × 10^4^
	Week 1 (*N* = 7)	4.60 × 10^4^	3.33 × 10^3^	5.48 × 10^4^	5.93 × 10^4^
	Week 2 (*N* = 3)	ND	7.02 × 10^2^	ND	ND
Mild to Moderate	Admission (*N* = 60)	9.35 × 10^7^	1.24 × 10^8^	2.45 × 10^7^	3.27 × 10^7^
	Week 1 (*N* = 55)	8.25 × 10^6^	9.11 × 10^6^	1.22 × 10^7^	2.20 × 10^7^
	Week 2 (*N* = 29)	8.77 × 10^5^	7.18 × 10^5^	1.15 × 10^6^	1.44 × 10^6^
Severe	Admission (*N* = 8)	5.70 × 10^6^	1.86 × 10^7^	2.86 × 10^6^	4.72 × 10^6^
	Week 1 (*N* = 8)	8.45 × 10^6^	8.78 × 10^6^	1.93 × 10^6^	1.34 × 10^6^
	Week 2 (*N* = 8)	3.18 × 10^5^	3.52 × 10^5^	2.21 × 10^6^	3.16 × 10^6^
Critical or Fatal	Admission (*N* = 18)	3.42 × 10^8^	3.42 × 10^8^	6.15 × 10^7^	1.01 × 10^8^
	Week 1 (*N* = 17)	1.61 × 10^6^	1.61 × 10^6^	8.35 × 10^6^	1.16 × 10^7^
	Week 2 (*N* = 14)	4.99 × 10^5^	7.13 × 10^5^	2.37 × 10^6^	2.61 × 10^6^

ND: not detectable; *N*: number of patients; mean cycle threshold values obtained under standard and viral loads are presented here. Cycle threshold >40 was considered positive according to the instructions of the Kogene Kit and SD Kit.

**Table 5 viruses-15-01560-t005:** Univariate logistic regression analysis of predictive risk factors for RNAemia presence.

Clinical Attributes	Univariate Logistic Regression Analysis		Multiple Logistic Regression Analysis	
	Odds Ratio (95% CI)	*p*-Value *	Odds Ratio (95% CI)	*p*-Value
**Age**	1.201 (1.078–1.338)	0.001	1.305 (1.061–1.604)	0.012
**Male**	0.732 (0.215–2.492)	0.617		
**RNAemia**	41.111 (7.751–218.044)	<0.001	17.301 (1.786–167.551)	0.014
**Upper RT viral *E* gene copy number**	1 (1–1)	0.076		
**Upper RT viral *RdRP* gene copy number**	1 (1–1)	0.065		
**Lower RT viral *E* gene copy number**	1 (1–1)	0.532		
**Lower RT viral *RdRP* gene copy number**	1 (1–1)	0.517		
**WBC × 10^9^ per L**	1.153 (0.982–1.354)	0.083		
**Neutrophil (%)**	1.085 (1.026–1.147)	0.004		
**Lymphocyte (%)**	0.905 (0.840–0.975)	0.009		
**Neutrophil/Lymphocyte ratio**	1.111 (1.031–1.197)	0.006	1.108 (0.901–1.363)	0.330
**C-reactive protein (mg/dL)**	1.161 (1.060–1.271)	0.001	1.083 (0.899–1.305)	1.083
**Troponin-I (ng/mL)**	126.079 (0.177–89,663.520)	0.149		
**Aspartate aminotransferase (U/L)**	1.023 (1.004–1.042)	0.018	1.039 (0.989–1.090)	0.126
**Alanine aminotransferase (U/L)**	0.986 (0.943–1.031)	0.542		

* *p* < 0.05 is considered significant. CI, confidence interval; CRP, C-reactive protein; AST, aspartate aminotransferase; ALT, alanine aminotransferase.

## Data Availability

Dong-Min Kim had full access to all data in the study and take responsibility for the integrity of the data and the accuracy of the data analysis. All data are available from the corresponding author upon reasonable request.

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
