# Peer review of "SARS-CoV-2 RNAemia and Disease Severity in COVID-19 Patients"

_viruses, 2023, doi:10.3390/v15071560_

Round 1

Reviewer 1 Report (New Reviewer)

Dear Editor,

Panchali and co-workers demonstrated that SARS-CoV-2 RNA viremia is a predictive risk factor for clinical severity in patients with COVID-19 patients. Blood RNAemia may might be a critical marker for disease severity and mortality. It was difficult to read the paper in the pdf format with the tracking of corrections. The scientific soundness is high. I have few comments for the authors:

- Introduction section needs to be improved, including more studies about the same field.

- Material and Methods section: it should be interesting to know the correlation between SARS-CoV-2 variant and RNAemia. Are the viral classification of viruses responsible of patient infection available? 

- Results section: the Figure 1 is not clear. Please add more details in the figure legend, for instance to specify the number in round brackets.

Results section: Cycle threshold (Ct) values were not compared to copy number in blood and respiratory samples. However, it should be useful to know this information. Especially, because the majority part of labs are not able to do a quantitative assays. 

- Discussion section: in Table 1 the authors referred about antiviral therapy, anyhow the effect of drug on viremia (according to time) is not discussed.

Please, add it.

Kind regards

Minor editing of English language required

Author Response

Response to Reviewer 1 Comments

Panchali and co-workers demonstrated that SARS-CoV-2 RNA viremia is a predictive risk factor for clinical severity in patients with COVID-19 patients. Blood RNAemia may might be a critical marker for disease severity and mortality. It was difficult to read the paper in the pdf format with the tracking of corrections. The scientific soundness is high. I have few comments for the authors:

Response: Thank you for your thorough and constructive comments towards our manuscript. We believe that the provided comments and suggestions have improved the quality of our manuscript. We are sorry for the track change version as it’s the revised version and as per journal policy we had to submit with the track changes, and the editorial team also made some significant changes.

  • Point 1: Introduction section needs to be improved, including more studies about the same field

Response 1: Thank you for your inputs we had added few more recent references and their findings in the introduction parts. Please refer to our revised manuscript.

  • Point 2: Material and Methods section: it should be interesting to know the correlation between SARS-CoV-2 variant and RNAemia. Are the viral classification of viruses responsible of patient infection available?

Response 2: Thank you for your valuable comments, unfortunately as the study was from the first wave of COVID19 we were unable to proceed with the viral load with variants study. It need further study with enlarged data.

  • Point 3: Results section: the Figure 1 is not clear. Please add more details in the figure legend, for instance to specify the number in round brackets.

Response 3: Thank you for the suggestion, we had updated the Figure legend if Fig 1, please refer to our revised Manuscript.

  • Point 4: Results section: Cycle threshold (Ct) values were not compared to copy number in blood and respiratory samples. However, it should be useful to know this information. Especially, because the majority part of labs are not able to do a quantitative assays.

Response 4: Thank you for raising the question, for your clarification as you can see from our supplementary figure we had already presented the date with the viral load correlation between respiratory samples vs RNAemia samples using the same Ct values. Our results confirms the significance of correlation of blood association with respiratory samples.

  • Point 5: Discussion section: in Table 1 the authors referred about antiviral therapy, anyhow the effect of drug on viremia (according to time) is not discussed.

Response 5: Thank you for your comments. We had added a sentence in the discussion part using the resent article published by Krifors et.al, were antiviral treatments doesn’t affect the viremia presence.

Reviewer 2 Report (New Reviewer)

Merlin Jayalal Lawrence Panchali and colleagues reported a study of the risk factor of SARS-CoV-2 RNA viral load in blood for COVID-19 severity. The authors investigated a cohort of 95 patients with confirmed COVID-19 positive diagnosis. The results showed a higher SARS-CoV-2 RNAemia (67.7%) in COVID-19 fatal cases compared to mild-moderate cases. Moreover, Viral RNAemia was detected in the 1st week 1 hospitalization samples but not in the samples during collected on the second week of hospitalization post-symptom onset. Statistical analysis showed that SARS-CoV-2 RNAemia was a predictive risk factors for COVID-19 severity. Collectively the study reports an interesting data that may have a clinical relevance. However, there are some point that should be improved.

Main points

1- In materials and methods is was reported that all samples were confirmed COVID-19 positive with molecular diagnosis and cell cultures. Since in results was reported that in none of positive SARS-CoV-2 RNA plasma sample SARS-CoV-2 was isolated, what type of samples was used in the first diagnosis (nasopharyngeal swab)? This should be indicated.

2- It is not clear the number of positive RNA samples. In table 1 there are indicated 19 patients/samples, in the text (line 197) it is reported 12 patients/samples.

3- The paragraph of cell-culture of SARS-CoV-2 RNAemia-positive…should be moved after that of Viral SARS-CoV-2 RNAemia…

4- Since the authors reported a positive isolation in cell culture in first diagnosis (lines 75-76) but none for positive palsma samples, it should be commented if this depend from the antibody present in plasma or other factors that impaired virus isolation using plasma/serum. Moreover, this can be related to the presence of viral genomic fragment and not viral whole particles in blood?

5- Coupled to the previous point, the negative result of sampling at 2 wks should be commented. Why the SARS-CoV-2 positivity is not present after 2 wks in patients with high SARS-CoV-2 yield in respiratory tract?

Minor points

1- In all the manuscript SARS-Cov-2 should be correct in SARS-CoV-2.

2-In all the manuscript “RNAemia” should be changed with “SARS-CoV-2 RNAemia”.

3- Figure 1: the title of the axis is lack.

4- Table 2: the title could be changed in “Biochemical and immunological findings of patients with COVID-19 on admission”.

5- Figure 1 and 2 should be improved and their legend should be more descriptive.

6- lines 207 and 210: the supplementary figure 1 should be added.

Author Response

Response to Reviewer 2 Comments

Merlin Jayalal Lawrence Panchali and colleagues reported a study of the risk factor of SARS-CoV-2 RNA viral load in blood for COVID-19 severity. The authors investigated a cohort of 95 patients with confirmed COVID-19 positive diagnosis. The results showed a higher SARS-CoV-2 RNAemia (67.7%) in COVID-19 fatal cases compared to mild-moderate cases. Moreover, Viral RNAemia was detected in the 1st week 1 hospitalization samples but not in the samples during collected on the second week of hospitalization post-symptom onset. Statistical analysis showed that SARS-CoV-2 RNAemia was a predictive risk factors for COVID-19 severity. Collectively the study reports an interesting data that may have a clinical relevance. However, there are some point that should be improved.

Response: Thank you for your constructive and encouraging comments towards our manuscript. We believe that the provided comments and suggestions have improved the quality of our manuscript.

Major Comments

  • Point 1: In materials and methods is was reported that all samples were confirmed COVID-19 positive with molecular diagnosis and cell cultures. Since in results was reported that in none of positive SARS-CoV-2 RNA plasma sample SARS-CoV-2 was isolated, what type of samples was used in the first diagnosis (nasopharyngeal swab)? This should be indicated.

Response 1: Thank you for your valuable comments. Yes as you mentioned the initial screening was made with respiratory samples, as per your suggestion we had added in our revised manuscript.

  • Point 2: It is not clear the number of positive RNA samples. In table 1 there are indicated 19 patients/samples, in the text (line 197) it is reported 12 patients/samples.

Response 2: Thank you for the comment, we apologize for the confusion, as from the MDPI correction they had removed the phrase “12 patients in critical and fatal category on admission we had corrected it in our revised manuscript.

  • Point 3: The paragraph of cell-culture of SARS-CoV-2 RNAemia-positive…should be moved after that of Viral SARS-CoV-2 RNAemia…

Response 3: Thank you for your valuable suggestion, we had moved the cell culture results after table 3.

  • Point 4: Since the authors reported a positive isolation in cell culture in first diagnosis (lines 75-76) but none for positive palsma samples, it should be commented if this depend from the antibody present in plasma or other factors that impaired virus isolation using plasma/serum. Moreover, this can be related to the presence of viral genomic fragment and not viral whole particles in blood?.

Response 5: Thank you for the valuable comment. In order to clarify it we used cell culture as one of the diagnostic method to confirm covid-19 patients. As per our results and previous published results, it’s not yet confirmed that the viral RNA is present is which form, most culture results including our results confirms that the culture results are negative with RNAemia samples. So how the virus or viral particles exist in blood is still inconclusive and need further studies.

  • Point 5: Coupled to the previous point, the negative result of sampling at 2 wks should be commented. Why the SARS-CoV-2 positivity is not present after 2 wks in patients with high SARS-CoV-2 yield in respiratory tract?

Response 5: Thank you for the comment, as you can see from the previous studies, the duration of RNAemia is less than 2 weeks and our results is also consistent with the previous studies. The reason behind this phenomenon is also still inconclusive and need further studies.

  • Point 5: Minor comments

â–¶ 1.  In all the manuscript SARS-Cov-2 should be correct in SARS-CoV-2.

Response 1: Thank you for the suggestion, we had updated the revised manuscript according to your suggestion.

â–¶ 2. In all the manuscript “RNAemia” should be changed with “SARS-CoV-2 RNAemia”.

Response 2: Thank you for the suggestion, we had updated the revised manuscript according to your suggestion.

â–¶ 3 Figure 1: the title of the axis is lack.

Response 3: Thank you for your suggestion, we had added fig1 axis details in our revised manuscript.

â–¶ 4 Table 2: the title could be changed in “Biochemical and immunological findings of patients with COVID-19 on admission”

Response 4: Thank you for your suggestion, we had chnged in our revised manuscript as you had suggested.

â–¶ 5 Figure 1 and 2 should be improved and their legend should be more descriptive.

Response 5: Thank you for your suggestion, we had added more information in figure legends as you had suggested.

â–¶ 5 lines 207 and 210: the supplementary figure 1 should be added.

Response 6: Thank you for your suggestion, however we apologize for regarding this comment, as you can see the lines you mentioned we present the data of RNAemia in blood and the sup Fig is of the correlation data of respiratory samples with blood and was well explained in separate secession after Fig. If we add in between the lines of 207 and 210 it may leads to a confusion for readers. Thank you for the understanding.

Round 2

Reviewer 2 Report (New Reviewer)

The authors have substantially improved the text. Now the manuscript is suitable for the pubblication.

This manuscript is a resubmission of an earlier submission. The following is a list of the peer review reports and author responses from that submission.

Round 1

Reviewer 1 Report

Abstract/objective: It should be clarified that the topic is SARS-Cov-2 RNA viremia. (The same is true for various passages in the manuscript).

The introduction seems rather outdated - with figures for the Covid-19 pandemic as of September 2021.

More details on the type and duration of antiviral therapies should be listed-especially since apparently all 19 patients with SARS-CoV-2 RNAemia (Table 1) were treated with antiviral drugs. This could also have an impact with regard to the lack of detection of SARS-Cov-2 RNA at week 2.

It would also be interesting to know whether patients with immunodeficiency were included - this could relevantly influence the extent and duration of viremia.

line 169/170 and 179, Table 1 and 3: Inconsistent numbers regarding the amount of patients with SARS-CoV-2 RNAemia - in line 169/170 and 179 12 patients with SARS-Cov-2 viremia are mentioned, this is inconsistent with the number of 19 patients in Table 1 and 14 patients with positive RNA at admission in Table 3. line 174: In contrast, the statement of 2% positive SARS-Cov-2 RNA on admission is more consistent with the figure of 19.

Principally interesting study on SARS-COV-2 viremia and its clinical relevance for disease progression.

However, due to inconsistency of the data in the current state unsatisfactory in terms of scientific content.

Author Response

Thank you for your thorough and constructive comments towards our manuscript. We believe that the provided comments and suggestions have improved the quality of our manuscript.

  • Point 1: Abstract/objective: It should be clarified that the topic is SARS-Cov-2 RNA viremia. (The same is true for various passages in the manuscript).

Response 1: Thank you for the comment.  In our study, if we use SARS-CoV-2 RNA viremia, the terminology may lead to certain confusion. As our cell culture results of SARS-CoV-2 RNA virus from patients’ blood is negative, so RNAemia seems to be more adequate. Therefore, we just used RNAemia.

  • Point 2: The introduction seems rather outdated - with figures for the Covid-19 pandemic as of September 2021.

Response 2: Thank you for your inputs we had added few more recent references and their findings in the introduction parts.

  • Point 3: More details on the type and duration of antiviral therapies should be listed-especially since apparently all 19 patients with SARS-CoV-2 RNAemia (Table 1) were treated with antiviral drugs. This could also have an impact with regard to the lack of detection of SARS-Cov-2 RNA at week 2.

Response 3: Thank you for your comment. According to the recent published manuscript https://doi.org/10.1371/journal.pone.0281052, the RNAemia percentage varies from 1st wave to 4th wave of SARS CoV2 infections. In our study a total of 19 patients had RNAemia, of them 17 were administered with remdesivir for an average of 4 days. Two patients used Kaletra for an average of 9 days. Moreover, another study “PLoS One. 2021 Jul 13;16(7):e0254640. doi: 10.1371/journal.pone.0254640.” used remdesivir and studied the effect of antiviral treatment with RNAemia patients and there was no significant correlation with RNAemia and antiviral treatment. Our results is also consistent with the study and we believe that antiviral treatments may not affect the presence of RNAemia. We had updated our table 1 with the antiviral treatments in our revised manuscript.

  • Point 4: It would also be interesting to know whether patients with immunodeficiency were included - this could relevantly influence the extent and duration of viremia.

Response 4: Thank you for raising the question, in our study we had 19 RNAemia patients and none of them are in immunosuppressant condition including HIV during the study period. (In the severe group, there was one brain cancer patient, and in the critical group, there was one lung cancer patient. None of these patients were receiving chemotherapy.)

  • Point 5: line 169/170 and 179, Table 1 and 3: Inconsistent numbers regarding the amount of patients with SARS-CoV-2 RNAemia - in line 169/170 and 179 12 patients with SARS-Cov-2 viremia are mentioned, this is inconsistent with the number of 19 patients in Table 1 and 14 patients with positive RNA at admission in Table 3. line 174: In contrast, the statement of 2% positive SARS-Cov-2 RNA on admission is more consistent with the figure of 19.

Response 5: Thank you for your comments, we apologize if you got some misunderstanding in our data. We would like to clarify your comments. The total patients who had RNAemia is 19, however the numbers which you mentioned are according to the classification of patients and the presence of RNAemia in accordance with time period, So the data presented in Table 1 and 3 are different. Hope we clarified your query.

  • Point 6: Principally interesting study on SARS-COV-2 viremia and its clinical relevance for disease progression.

Response 6: Thank you for your encouraging comments.

  • Point 7: However, due to inconsistency of the data in the current state unsatisfactory in terms of scientific content.

Response 7: Thank you for the constructive comments and criticisms of our manuscript named above. We believe that your valuable comments and by making those changes our manuscript has been considerably improved

Reviewer 2 Report

In this manuscript, the authors investigated the dynamics of SARS-CoV-2 viral RNA detected in peripheral blood samples of COVID-19 patients. To carry out the cohort, a total of 95 patients with confirmed COVID-19 were selected, which are representative of the period from February/2020 to May/2021 at Chosun University Hospital, South Korea. In summary, to achieve the proposed objectives, the authors used qRT-PCR, cell culture, ELISA, and IFA for SARS-CoV-2 detection techniques. The main result was related to proportion of viral RNAemia was directly correlated with disease severity.

Major Comments

My biggest concern is related to the low n sample included in the study (n=95) for a disease that in the years 2020-2022 was highly prevalent in several regions of the world. Thus, the robustness of the data is something that can generate conclusion bias related to the higher levels of SARS-CoV-2 RNAemia associated with the clinical severity of COVID-19. Although the studies cited by the authors also used a small sample size, it should be noted that they were published at the beginning of the COVID-19 pandemic. Therefore, I think it is important for the authors to mention the limitations of their study, including discussing more about the possible effect of comorbidity on viral load and its results.

Introduction and Discussion: These sections could be improved due to its paucity of discussions.

Minor comments

1. Line 46: It is necessary to update the epidemiological data with the number of confirmed cases and deaths from COVID-19 for the year 2023.

2. Line 71: Figure 01 should be before or after Table 3.

3. Line 222. Table 05: As previously mentioned, the low number of samples and heterogeneity contributed to an OR with a very wide range of variance (i.e., RNAemia= 41·111(7·751–218·044). I see it as important to discuss this population heterogeneity that seems to exist.

Author Response

In this manuscript, the authors investigated the dynamics of SARS-CoV-2 viral RNA detected in peripheral blood samples of COVID-19 patients. To carry out the cohort, a total of 95 patients with confirmed COVID-19 were selected, which are representative of the period from February/2020 to May/2021 at Chosun University Hospital, South Korea. In summary, to achieve the proposed objectives, the authors used qRT-PCR, cell culture, ELISA, and IFA for SARS-CoV-2 detection techniques. The main result was related to proportion of viral RNAemia was directly correlated with disease severity.

Major Comments

  • Point 1: My biggest concern is related to the low n sample included in the study (n=95) for a disease that in the years 2020-2022 was highly prevalent in several regions of the world. Thus, the robustness of the data is something that can generate conclusion bias related to the higher levels of SARS-CoV-2 RNAemia associated with the clinical severity of COVID-19. Although the studies cited by the authors also used a small sample size, it should be noted that they were published at the beginning of the COVID-19 pandemic. Therefore, I think it is important for the authors to mention the limitations of their study, including discussing more about the possible effect of comorbidity on viral load and its results.

Response 1: Thank you for your encouraging comments, we had added the information’s as you mentioned in the limitation of this study in our revised manuscript.

The possible effect of viral load due to underlying comorbidity need to be further studied.

Please refer to revised manuscript.

  • Point 2: Introduction and Discussion: These sections could be improved due to its paucity of discussions.

Response 2: Thank you for the comment, we had updated the introduction and discussion. please refer to our revised manuscript.

  • Point 3: Minor comments

â–¶ 1. Line 46: It is necessary to update the epidemiological data with the number of confirmed cases and deaths from COVID-19 for the year 2023.

Response 3: Thank you for the suggestion, we had updated the numbers according to the present situation.

â–¶ 2. Line 71: Figure 01 should be before or after Table 3.

Response 3: According to our data presented here the Figure 1 will be first followed by Table 3. Thank you.

â–¶ 3. Line 222. Table 05: As previously mentioned, the low number of samples and heterogeneity contributed to an OR with a very wide range of variance (i.e., RNAemia= 41·111(7·751–218·044). I see it as important to discuss this population heterogeneity that seems to exist.

Response 3: Thank you for your suggestion, we had added a discussion point in our revised manuscript as you had suggested.

Reviewer 3 Report

Authors showed that “our results confirmed the presence of RNA in blood samples during the initial stage of infection, especially in the older population, which might be a marker of disease severity”

- Do authors have any idea how this virus exists in blood? If authors have it, could you comment about this in the discussion part? 

- Do authors have the relationship between cytokine profile especially cytokine storm related cytokines and RNA in blood (RNAemia)? Because authors showed that “the presence of RNAemia in critical or fatal cases was the highest (66.7%)”. 

- Can RNA in blood (RNAemia) transmit to a healthy person? 

Author Response

Thank you for your thorough and constructive comments towards our manuscript. We believe that the provided comments and suggestions have improved the quality of our manuscript.

  • Authors showed that “our results confirmed the presence of RNA in blood samples during the initial stage of infection, especially in the older population, which might be a marker of disease severity”. Do authors have any idea how this virus exists in blood? If authors have it, could you comment about this in the discussion part? 

Response 1: Thank you for your comment. As per our results and previous published results, it’s not yet confirmed that the viral RNA is present is which form, most culture results including our results confirms that the culture results are negative. So how the virus or viral particles exist in blood is still inconclusive and further study is needed to find the existence of virus. We had added a point in the revised manuscript.

  • Do authors have the relationship between cytokine profile especially cytokine storm related cytokines and RNA in blood (RNAemia)? Because authors showed that “the presence of RNAemia in critical or fatal cases was the highest (66.7%)”. 

Response 2: Thank you for your comments, unfortunately as the study was from the first wave of COVID19 we were unable to proceed with the cytokine study. It need further study with cytokine data. We have added that point in our revised manuscript

  • Can RNA in blood (RNAemia) transmit to a healthy person? 

Response 3: Thank you for raising the question. As the blood is having the least viral detection and most culture results are negative, so the possibility of infection by blood is very low even though it is inconclusive and need further study for confirmation.

Round 2

Reviewer 1 Report

In regard to response 4:

Since you clarified that the total number of patients in whom SARS-CoV RNA was detected in blood was 19, the 12 patients analyzed in cell cultures mentioned in lines 198 and 210 refer to the critical/fatal category at admission, correct? This should be clearly mentioned in these paragraphs, otherwise there may be also misunderstanding by the reader.

Author Response

Thank you for your input, we had reframed the sentense as below for better understanding in our revised manuscript. 

"None of the 12 patients in critical and fatal category on admission with RNAemia showed positive culture results."

Reviewer 2 Report

My suggestion is just to correct lines 98 and 99, where the phrase 'primers and probes' are in duplicate. Perform an analysis verifying that there are no other textual errors of this type in the manuscript.

Author Response

Thank you for your inputs we had deleted the repeated phrases. In addition we rechecked the manuscript to check weather there are any errors.